# Irradiation Activates MZF1 to Inhibit miR-541-5p Expression and Promote Epithelial-Mesenchymal Transition (EMT) in Radiation-Induced Pulmonary Fibrosis (RIPF) by Upregulating Slug

**DOI:** 10.3390/ijms222111309

**Published:** 2021-10-20

**Authors:** Xinxin Liang, Ziyan Yan, Ping Wang, Yuhao Liu, Xingkun Ao, Zheng Liu, Duo Wang, Xiaochang Liu, Maoxiang Zhu, Shanshan Gao, Dafei Xie, Pingkun Zhou, Yongqing Gu

**Affiliations:** 1Hengyang Medical College, University of South China, Hengyang 421001, China; axinl96@163.com (X.L.); aoxingkun@163.com (X.A.); 2Beijing Key Laboratory for Radiobiology, Beijing Institute of Radiation Medicine, Beijing 100850, China; yanziyan777@163.com (Z.Y.); wp17803901673@163.com (P.W.); yuhaoliu97@163.com (Y.L.); 15829309917@163.com (D.W.); lxiaochang@163.com (X.L.); zhumx@nic.bmi.ac.cn (M.Z.); gaoshanbprc@163.com (S.G.); 15892752837@163.com (D.X.); 3School of Public Health, University of South China, Hengyang 421001, China; l17802303666@163.com

**Keywords:** miR-541-5p, EMT, slug, MZF1, radiation-induced pulmonary fibrosis

## Abstract

Understanding miRNAs regulatory roles in epithelial-mesenchymal transition (EMT) would help establish new avenues for further uncovering the mechanisms underlying radiation-induced pulmonary fibrosis (RIPF) and identifying preventative and therapeutic targets. Here, we demonstrated that miR-541-5p repression by Myeloid Zinc Finger 1 (*MZF1*) promotes radiation-induced EMT and RIPF. Irradiation could decrease miR-541-5p expression in vitro and in vivo and inversely correlated to RIPF development. Ectopic miR-541-5p expression suppressed radiation-induced-EMT in vitro and in vivo. Knockdown of Slug, the functional target of miR-541-5p, inhibited EMT induction by irradiation. The upregulation of transcription factor *MZF1* upon irradiation inhibited the expression of endogenous miR-541-5p and its primary precursor (pri-miR-541-5p), which regulated the effect of the *Slug* on the EMT process. Our finding showed that ectopic miR-541-5p expression mitigated RIPF in mice by targeting *Slug*. Thus, irradiation activates *MZF1* to downregulate miR-541-5p in alveolar epithelial cells, promoting EMT and contributing to RIPF by targeting *Slug*. Our observation provides further understanding of the development of RIPF and determines potential preventative and therapeutic targets.

## 1. Introduction

Radiation-induced pulmonary fibrosis (RIPF) is one of the most serious complications in patients undergoing chest tumor radiotherapy, generally occurring at later stages of radiation therapy [1,2]. Fibroblasts are the key cells in pulmonary fibrosis. It is reported that over 30% of fibroblasts in pulmonary fibrosis models originate from the epithelial-mesenchymal transition (EMT) [3]. EMT is a cellular process that transforms epithelial to mesenchymal cells and gains cell movement. The exact mechanism underlying the association among IR (ionizing radiation), EMT, and RIPF remains to be elucidated to date.

MicroRNAs (miRNAs) are known to regulate gene expression by binding to the 3′-untranslated region (3′-UTR) of the relevant target messenger RNA (mRNA) [4,5,6,7]. Thus, the role of microRNAs in the development of RIPF should not be neglected. In our research, miR-541-5p was found down-regulated in the radiation-induced pulmonary fibrosis model. MiR-541 was reported that the dysregulation of the miR-541-*ATG2A/RAB1B* axis plays a key role in determining the patient’s response to sorafenib treatment [8]. Moreover, miR-541 could inhibit proliferation and migration in osteosarcoma cells and prostate cancer (PCa) cells [9,10]. Although miR-541-5p research is of great significance, its role in the RIPF research mechanism is unclear.

*Slug (Snai2)* is essential for the development of EMT and is well-documented to promote EMT via the inhibition of E-cadherin transcription [11,12]. In prostate cancer (PCa), Slug plays an important role in the EMT of PCa as a direct effector of miR-3622a [13]. In another report, the oncofetal IGF2 mRNA-binding protein 1 (*IGF2BP1*) was observed to force the synthesis of the “EMT-driving” transcriptional regulator *Slug* by promoting the expression of *LEF1*(Lymphoid enhancer-binding factor 1), which regulates EMT [14]. Therefore, it becomes important to study the regulatory mechanism underlying the role of *Slug* in EMT development. Studies on the involvement of *Slug* in major diseases are abundant, although the exact mechanism of IR regulation by *Slug* in RIPF remains unknown to date.

Moreover, the present study involved deciphering the mechanisms through which miR-541-5p is regulated in response to IR. The results revealed the role of the transcription factor *MZF1* (Myeloid Zinc Finger 1), which has been previously reported to be an important transcriptional repressor. *MZF1* is reported to repress the expression of the chloramphenicol acetyltransferase (CAT) reporter gene via *GAL4* (the yeast transactivator) binding sites in the non-hematopoietic cell lines NIH 3T3 and 293 [15]. Moreover, in NPM-ALK^+^ (nucleophosmin-anaplastic lymphoma kinase) T-cell lymphoma, *MZF1* reduces IGF-IR (the type I insulin-like growth factor receptor) expression by inhibiting its transcription [16].

Our data showed that IR could activate *MZF1*, thereby inhibiting the production of pri-miR-541-5p and reducing mature miR-541-5p, which would otherwise increase the *Slug* levels and promote the EMT process, resulting in a severe consequence of RIPF. This study provides further understanding of the development of RIPF and determines potential preventative and therapeutic targets.

## 2. Results

### 2.1. Ionizing Radiation Can Reduce the Expression of miR-541-5p

According to our previous report, irradiation could cause the occurrence of EMT and lead to changes in the content of miRNAs in both A549 and BEAS-2B cells [17,18]. RT-qPCR was employed to detect the levels of miR-541-5p at 0 h, 3 h, 6 h, and 48 h after the irradiation. As depicted in Figure 1A,B, the levels of miR-541-5p had decreased significantly after irradiation. These results were verified (Figure 1C) in a *mouse* model of RIPF (25 Gy, chest irradiation; lung tissue extracted at 1, 2, 3, and 4 months after the irradiation).

### 2.2. Knockdown of miR-541-5p Could Promote the Occurrence of EMT

Our previous results showed that miR-541-5p expression is down-regulated after irradiation, so we hypothesized that miR-541-5p might play a crucial role in EMT. Inhibitor of miR-541-5p was transfected into A549 cells and BEAS-2B cells, and RT-qPCR was performed to determine cell transfection efficiency (Figure 2A–D, left). Western blot experiments were performed to verify the changes in the epithelial-mesenchymal marker (Figure 2A–D, right). The results indicated that the epithelial marker E-cadherin was significantly decreased, while the mesenchymal markers N-cadherin and Vimentin were significantly increased when miR-541-5p was knocked down. Next, miR-541-5p was overexpressed after IR. As depicted in Figure 2B,D, it was found that over-expression of miR-541-5p can negatively regulate the EMT process induced by irradiation, and the corresponding protein markers showed changes opposite to those in the single irradiation group. This suggested that miR-541-5p overexpression could effectively inhibit the EMT process induced by IR.

### 2.3. miR-541-5p Directly Suppressed Slug via Binding to the 3′-UTR Region

To find miR-541-5p target genes, an online database (TargetScan; http://www.targetscan.org/vert_71/ accessed on 18 October 2021) was used to predict the target genes and possible binding sites that might be regulated by miR-541-5p. *Slug* was identified, and subsequently, its levels were determined by RT-qPCR and Western blot. The results revealed that both mRNA and protein levels of *Slug* were significantly elevated after the irradiation in A549 and BEAS-2B cells (Figure 3A,B). In order to better demonstrate the *Slug*-related changes in pulmonary fibrosis, *mouse* (GSE85359) and *human* (GSE40839) pulmonary fibrosis datasets were down-loaded from the NCBI (https://www.ncbi.nlm.nih.gov/ accessed on 18 October 2021). It was revealed that Slug expression was increased in both the fibrotic lung tissues (Figure 3C). Then, miR-541-5p mimic and inhibitor were transfected into A549 cells. The results showed that *Slug* expression was significantly decreased upon miR-541-5p overexpression, while it was significantly increased upon miR-541-5p knockdown. These results were verified at both mRNA level and protein level (Figure 3D,E). We predicted the binding sites of miR-541-5p and *Slug* and constructed the wild-type and mutant plasmids of *Slug* (Figure 3F). The results of the dual-luciferase reporter assay revealed that upon cotransfecting the miR-541-5p mimic with the WT plasmid of Slug into the HEK-293T cells, the fluorescence was significantly reduced. In contrast, upon co-transfection with the mimic, the fluorescence of the *Slug*’s MUT plasmid group was not significantly different from that of the normal group (Figure 3G). This finding indicated that miR-541-5p could directly target and regulate *Slug*.

### 2.4. miR-541-5p Negatively Regulated EMT by Inhibiting Slug

Although we demonstrated that miR-541-5p directly targets *Slug*, an essential protein of EMT, the relationship between miR-541-5p, *Slug*, and IR-induced EMT has not been elucidated. Next, siSlug was transfected post-IR or cotransfected with miR-541-5p inhibitor to observe the changes of EMT-related proteins. The protein expression levels of E-cadherin decreased, and N-cadherin, Vimentin elevated in IR groups. In contrast, the protein expression of E-cadherin increased, and N-cadherin, Vimentin decreased in IR+siSlug groups (Figure 4A). We knocked down *Slug* in cells with low miR-541-5p expression and observed the changes of related proteins of EMT (Figure 4B). In our data, the EMT process was inhibited when we continued to knock out *Slug* in cells with low miR-541-5p expression. Moreover, an immunofluorescent staining assay was used to confirm our findings. Red fluorescence indicated E-cadherin, and green fluorescence indicated N-cadherin. The changing trends of E-cadherin and N-cadherin confirmed the same trend as the protein levels (Figure 4C,E). In addition, a quantitative analysis of the area and the relative fluorescence was performed using ImageJ software (Figure 4D,F). Thus, the siSlug can negatively regulate EMT induced by IR and knockdown of miR-541-5p.

### 2.5. IR Downgrades miR-541-5p via MZF1

To determine whether miR-541-5p accumulation is mediated by transcriptional regulation, the expression of the primary precursor of miR-541-5p (pri-miR-541-5p) was evaluated. It was revealed that the pri-miR-541-5p levels were significantly reduced after irradiation in both A549 and BEAS-2B cells (Figure 5A). Therefore, it was inferred that IR reduced the levels of mature miR-541-5p by decreasing the production of pri-miR-541-5p. To understand how IR regulates the changes in miR-541-5p, the key transcription factors that could regulate the transcription of miR-541-5p were explored. The online database JASPAR (http://jaspar.genereg.net/ accessed on 18 October 2021) was used for predicting the transcription factors that could bind to the promoter region of miR-541-5p. In the process of searching, *MZF1*, a transcriptional suppressor, was found to have a binding site in the promoter region of miR-541-5p. In order to determine the relationship between IR, *MZF1*, and miR-541-5p, we first examined whether *MZF1* increased after irradiation. Our results showed that the *MZF1* expression increased significantly in A549 and BEAS-2B cells after irradiation (Figure 5B,C). Furthermore, to understand the dynamics of *MZF1* expression in the lung fibrosis tissue, the GSE85359 (*Mouse*) and GSE40839 (*Human*) datasets were downloaded from NCBI. Both datasets revealed increased expression of MZF1 (Figure 5D). Next, overexpression or siRNA was used to raise or knock down *MZF1*′s expression specifically. RT-qPCR results revealed that overexpression of *MZF1* inhibits the expressions of pri-miR-541-5p and miR-541-5p (Figure 5E,F), indicating that *MZF1* could reduce the production of mature miR-541-5p by affecting pri-miR-541-5p.

Here, we showed the predicted binding sites of the miR-541-5p promoter to *MZF1* and constructed wild-type and mutant plasmids (Figure 5G). The luciferase assay revealed that the fluorescence activity of the wild-type miR-541-5p promoter was significantly reduced upon *MZF1* overexpression, while little change occurred in the control (Figure 5H). All the above experiments fully proved that the expression of miR-541-5p was regulated by transcription factors *MZF1* under irradiation.

### 2.6. MZF1 Promotes EMT via Repression of miR-541-5p Following IR

Next, we attempted to follow up whether knockdown *MZF1* affected IR-induced EMT. As seen in Figure 6A, we demonstrated with Western Blot that when *MZF1* was overexpressed in cells, there was a change in the EMT-related proteins after 48 h. E-cadherin was significantly decreased, while the N-cadherin and Vimentin were significantly increased. The expression of E-cadherin increased, and the content of N-cadherin and Vimentin decreased compared with the irradiation group. These results suggested that when we overexpress *MZF1*, the EMT process is activated; thus, knockdown *MZF1* can negatively regulate the EMT induced by irradiation (Figure 6B).

Given that *MZF1* had an inhibitory effect on miR-541-5p expression, we transfected *MZF1* overexpression plasmid, and miR-541-5p mimic into cells and observed the changes in the EMT-related proteins. Reliable experimental results demonstrated that overexpressed both *MZF1* and miR-541-5p mimic had higher levels of E-cadherin and significantly lower levels of mesenchymal markers than those that overexpressed only *MZF1* (Figure 6C). Thus, suggesting that increasing the level of miR-541-5p can effectively relieve the EMT process in the presence of over-expression of *MZF1*. Since *MZF1* could inhibit miR-541-5p, which could target Slug, we explored what happens to EMT-associated proteins when *MZF1* is overexpressed, and Slug is knocked down. The Western blot experiments revealed that *Slug* knockdown in cells could effectively inhibit the EMT process induced by *MZF1* (Figure 6D), suggesting that when we overexpressed *MZF1*, whether we increase miR-541-5p or knock down its target gene *Slug*, we inhibit the process caused by the overexpression of *MZF1*. The above results demonstrated that *MZF1* could stimulate the EMT process via the *MZF1*/miR-541-5p/*Slug* signaling axis following IR.

### 2.7. MZF1 Mediates RIPF via miR-541-5p/Slug Axis

A total of 80 mice were randomly divided into four groups of 20 mice each (Figure 7A)—Con group, IR group, IR+NC group, and IR+miR-541-5p mimic group (IR+mimic). An AAV vector was used for delivering the miR-541-5p mimic specifically to the lungs of the mice through the special administration. The lung tissues were retrieved from the mice at the first, second, third, and fourth months after the irradiation, and the levels of miR-541-5p in the mouse lung tissues were evaluated. Compared with the unirradiated group, the miR-541-5p contents were significantly lower in the IR group and IR+NC group. In contrast, the IR+mimic group presented high expression levels of miR-541-5p in the lung tissue even after four months (Figure 7B). Next, the results of H&E staining of the mouse lung tissues were examined and scored. It is evident from the images (Figure 7C) that the alveolar tissue of irradiated mice was destroyed, and the alveolar structure showed incomplete morphology, such as rupture with the prolongation of irradiation time. In addition, the alveolar wall was significantly thickened, the alveolar septum was significantly widened, and the dense degree of lung tissue was much greater than the unirradiated group. These findings illustrated that the RIPF model had been successfully established.

In comparison to the IR and IR+NC groups, the IR+mimic group presented a significant reduction in the severity of RIPF. The same results were obtained when scoring the radiation-induced pulmonary fibrosis (Figure 7D). In the lung tissue samples of the Con group, IR could significantly cause fibrotic lesions, which were largely improved upon miR-541-5p overexpression. In order to observe the collagen deposition in mouse lungs, Masson staining of the mouse lung tissues was performed (Figure 7E). In the figure, blue represents the collagen, which was quantified using the ImageJ software (Figure 7F). The staining results and the quantitative results collectively indicated that there was severe collagen deposition in the IR and IR+NC groups, which became further evident with time. Thus, it indicated that the collagen production and accumulation in the lungs increased with the time after irradiation. This is an important part of the RIPF formation process. However, for the lung tissue of mice in the miR-541-5p high expression group, both the denseness of the alveoli and the thickness of the alveolar septum significantly reduced compared with the irradiated group. Collagen deposition was also lower than the irradiated group. Thus, it suggested that when we overexpress miR-541-5p in vivo, it can significantly protect the lung’s structural damage caused by IR and significantly reduce the IR-induced RIPF. Furthermore, proteins were also extracted from the mouse lung tissues to quantify the amounts of Collagen I and the waveform protein α-SMA. It was revealed that at the fourth month, the levels of Collagen I and α-SMA were significantly increased in the lung tissue from the IR and IR+NC groups, while the levels of these proteins in the IR+mimic group were significantly lower compared to the previous two groups (Figure 7G). It is also proved that the reversal effect of miR-541-5p on the IR-induced RIPF process from the side. Next, the levels of *MZF1* and target *Slug* gene in mice were also evaluated at the fourth month. *MZF1* levels presented a significant increase after irradiation, while *Slug* increased after irradiation but decreased in the IR+mimic group (Figure 7H). Simultaneously, immunohistochemical was employed to evaluate *Slug* expression in mouse lung tissue (Figure 7I,J). This suggested that miR-541-5p overexpression in mice was accompanied by a decrease in *Slug* levels, which is consistent with the results obtained in the in vitro experiments of the present study. Therefore, it was inferred that miR-541-5p overexpression in mice significantly ameliorated the lung lesions caused by RIPF and collagen deposition in mice in terms of both lung structure and collagen deposition results. The above experiments confirmed the therapeutic effect of miR-541-5p in RIPF. These findings combined with the in vitro experiment results demonstrate that *MZF1* mediates IR-induced pulmonary fibrosis via the miR-541-5p/*Slug* axis—this mechanism is illustrated in Figure 8.

## 3. Discussion

The development of radiation-induced pulmonary fibrosis (RIPF) through activating epithelial-mesenchymal transition (EMT) is usually complex and involves multiple molecules and genes. However, little information is available on how IR regulates the important transcription factor *Slug* of EMT. MicroRNAs (miRNAs) are considered important factors in developing various diseases, and research on the clinical therapeutic effects of miRNAs has never ceased. This study provides sufficient evidence that miR-541-5p can inhibit EMT induced by irradiation in vitro experiments. We also clearly observed that miR-541-5p can effectively interfere with RIPF formation in vivo experiments.

As a widespread and significant class of biological genes, miRNAs are reported to be inextricably linked to radiation-induced lung injury [19,20,21]. Previous studies have reported miR-541-5p as a key effector in lung fibroblasts, which influences bleomycin-induced pulmonary fibrosis by regulating the target gene *PDE1A* (phosphodiesterase 1A) [22]. Furthermore, miR-541 suppression mediates the promotion of *HSP27* (heat shock protein 27) expression during heat stress, which ultimately leads to activation of autophagy, inhibition of the mitochondrial apoptotic pathway, and the malignant transformation of human bronchial epithelial cells [23]. These experimental findings indicate the significance of miR-541. Coinciding with our study, we found that miR-541-5p expression was downregulated in pulmonary epithelial cells after irradiation, causing activation of the EMT process by targeting Slug, leading to an increased number of fibroblasts, and enhanced cell proliferation and migration, finally causing RIPF. In previous experiments, AAV has been successfully used for delivering siRNA or the mimic inside mice [17,23,24,25]. In our Study, AAV was used to carry miR-541-5p mimic for administration in the lungs of mice. We demonstrated that the use of AAV for delivering the miR-541-5p mimic into the mouse body was quite effective and long-lasting, with the high expression in vivo continuing even until the fourth month. The lung fibrosis was significantly reduced in the mice after the intervention with miR-541-5p mimic compared with the irradiated group, and the alveolar septum and tissue denseness were reduced. Collagen deposition was also lower, indicating that miR-541-5p has a significant preventive and therapeutic effect on RIPF. This indicates the feasibility of using AAV to deliver an inhibitor or mimic in the context of the era of vigorous development of molecular targeted therapy. Moreover, we believe that miR-541-5p can play an important role in the future as a biomarker or a preventive or therapeutic drug.

As Slug has been studied more intensively, its regulated network has become clearer. *Slug* belongs to the *Snail* family [26,27]. *Slug* contains zinc finger structures and is a key regulator of the EMT process [28,29,30,31]. Studies have reported that the *Slug* protein inhibits the expression of the cell adhesion molecule E-cadherin [30,32,33]. High *Slug* expression in RIPF often implies accelerated onset and progression of IR-induced EMT, and consequently, accelerated progression of RIPF [34,35]. The present study comprehensively demonstrated that miR-541-5p could silence the *Slug* expression and inhibit the radiation-induced EMT process by binding to the 3′-UTR of *Slug*. The same conclusion was reached with the results of the in vivo experiments. The overexpression of miR-541-5p in mice resulted in a significant reduction in the symptoms of RIPF in mouse lungs, which could maintain an almost normal alveolar structure along with the alveolar septum exhibiting a mild widening. At the same time, the fibrous exudate and vascular stasis conditions were significantly better than those in the IR and IR+NC groups. Therefore, it could be inferred that miR-541-5p would play a significant role in the treatment of RIPF. The global vaccination drive against COVID-19 is gradually highlighting the role of AAV as an effective molecular target therapy agent [36]. Besides corroborating the effectiveness of the therapeutic approach of AAV, the present work also provides a solid experimental foundation for the use of miR-541-5p as a molecular target therapy agent in RIPF.

Although numerous studies have reported that miRNA expression is altered upon irradiation, studies exploring the mechanisms underlying these miRNA changes in response to irradiation are scarce. Recently, it has been reported that circular RNA can act as a sponge for miRNA to influence its function [37,38,39]. Thus, we attempted to look for links that might regulate miRNA production, starting from the miRNA production pathway. A study reported that IR could alter the transcription factors *ATF2*, *ELK1*, and *YY1* to regulate the transcription process of miR-320a [21]. In the present study, the characteristics of the promoter region of miR-541-5p were analyzed, and JASPAR (http://jaspar.genereg.net/ accessed on 18 October 2021) was used for predicting the possible transcription factor-binding domains in this promoter region. *MZF1*, a class of transcription repressors, was found to specifically recognize the promoter, thereby reducing the production of pri-miR-541-5p, and consequently, the production of mature miR-541-5p. *MZF1* belongs to the family of zinc-finger transcription factor proteins, which are involved in regulating the transcriptional processes during different developmental processes [40,41]. Most studies report *MZF1* as a potent transcriptional repressor [42,43,44]. The homeostatic disruption of *MZF1* is reported to promote the conversion of an invasive mesenchymal phenotype to a less-invasive epithelial phenotype [45]. In addition, *MZF1* is reported to upregulate N-cadherin expression and promote EMT [46]. The in vitro experiments revealed that IR could activate *MZF1* to repress the transcription of miR-541-5p, attenuate its silencing effect on the target *Slug* gene, and promote the EMT process, which is a key link in RIPF. An in vivo experiment of our study revealed that the symptoms of RIPF in mice were significantly reduced after treatment with miR-541-5p, which suggested that the therapeutic effect of miR-541-5p should not be underestimated. Although we showed that *MZF1* expression increased after irradiation, there is limited research on the relationship between *MZF1* and irradiation-induced diseases. Thus, we look forward to more mechanistic studies on the effects of *MZF1* after irradiation in the future. Our study is of great interest for the transcriptional regulation of miRNAs and the formation of the EMT process and RIPF.

There are some limitations to our study. Although our mouse RIPF model well summarizes the main pathological features and changes occurring in human RIPF, there are many physiological differences between mice and humans. Therefore, the intervention effect of miR-541-5p in human RIPF needs to be further evaluated. The prevention and treatment of RIPF continue to be a challenge to date. Therefore, further research on RIPF is warranted to understand this process better and develop effective preventive or therapeutic drugs for the benefit of patients.

## 4. Materials and Methods

### 4.1. Cell Culture

The human alveolar type II epithelial cancer cell line A549 and the human normal lung epithelial cell line BEAS-2B were purchased from the National Collection of Authenticated Cell Cultures. Genetic information for all cell lines could find in the Cellosaurus database (https://web.expasy.org/cellosaurus/ accessed on 18 October 2021). A549 and BEAS-2B cells were cultured and maintained in high-glucose Dulbecco’s Modified Eagle’s Medium (DMEM, SIGMA, Saint Louis, MO, USA) supplemented with 10% fetal bovine serum (FBS; catalog number FSP500, ExCell Bio, Shanghai, China) under incubation at 37 °C in a humidified atmosphere containing 5% CO_2_.

### 4.2. RNA Isolation, Reverse Transcription, and qRT-PCR

Total RNA was isolated from cells using TRIzol^TM^ (Ambion, Thermo Fisher Scientific, Waltham, MA, USA) and eluted in 20 µL of RNase/DNase-free buffer (Biomed, RA114-02, Beijing, China), and then stored at –80 °C until further analysis. RNA concentration and quality were assessed using the Nanodrop 2000c spectrophotometer (Thermo Fisher Scientific, Waltham, MA, USA). RNA reverse transcription was performed for both A549 and BEAS-2B cell lines following the instructions provided by the miRcute Plus miRNA First-Strand cDNA Synthesis Kit (TIANGEN BIOTECH, Beijing, China) and the ReverTra Ace qPCR RT Master Mix with gDNA Removal Kit (Toyobo, Large Edition, Japan), respectively. The Reverse Transcription System was carried by Applied Biosystems PCR (Ambion, Thermo Fisher Scientific, Waltham, MA, USA). RT-PCR was performed using either the miRcute Plus miRNA qPCR kit (SYBR Green) (TIANGEN BIOTECH, Beijing, China) or the THUNDERBIRDTM SYBR qPCR mix (Toyobo, Large Edition, Japan) following the manufacturer’s instructions with CFX96 Touch^TM^ Real-Time PCR Detection System (Bio-Rad, Hercules, CA, USA). The cycling conditions for RNA were as follows: initial denaturation at 95 °C for 1 min, followed by 40 cycles of 94 °C for 15 s, 60 °C for 60 s. The cycling conditions for miRNA were as follows: initial denaturation at 95 °C for 15 min, followed by 40 cycles of 94 °C for 20 s, 60 °C for 34 s. U6 was selected as the internal control for miRNA and β-actin was selected for mRNA. All the RT-qPCR primers are listed in Table 1. End of the reaction, 2^−∆∆Ct^ was used to analyze the data.

### 4.3. Irradiation and Transfection

The cells were irradiated with ^60^Co γ-rays at a dose rate of 80.74 cGy/min. The siRNA and the mimic/inhibitor used in the present study were designed and constructed by GenePharma company (GenePharma, Suzhou, China); the sequences are as follows: siMZF1 5′- CCAAGCCUUUCUCCAUUUUTT-3′; siSlug 5′-ACUACAGUCCAAGCUUUCATT-3′; miR-541-5p mimic Sense:5′-AAAGGAUUCUGCUGUCGGUCCCACU-3′, Antisense: 5′-UGGGACCGACAGCAGAAUCCUUUUU-3′; miR-541-5p inhibitor 5′- AAAGGAUUCUGCUGUCGGUCCCACU-3′. The overexpression plasmids were purchased from Fenghui Biologicals (accession ID: NM_003422), while the plasmids used for the dual-luciferase reporter assay were constructed by TSNGKE Biotech. Cat numbers are listed below: MUT-miR-541-5p promoter: Y0040634-8; WT-miR-541-5p promoter: Y0040634-7; WT-Slug: Y0040634-3; MUT-Slug: Y0040634-4. The concentration of siRNA/miR-541-5p inhibitor/miR-541-5p mimic used was 50nM and plasmid was 2ug/mL. All transfections were conducted using lipofectamine 2000 (Invitrogen, Carlsbad, CA, USA) according to the manufacturer’s instructions. The serum-containing culture medium was replaced 6 h after the transfection. The RNA or the protein was isolated 24 h or 48 h later, respectively, for subsequent experiments.

### 4.4. Mice and Mice Treatment

About 6–8-weeks-old C57BL/6 male mice were purchased from Vital River Laboratory Animal Co. (Beijing, China) and reared in a standard animal feeding environment. The mice were randomly divided into four groups: CON, IR, IR+AAV-NC (IR+NC), and IR+AAV-miR-541-5p mimic (IR+mimic), each containing 20 mice. Adeno-associated virus (AAV, 6.84 × 1010 vg/mouse, GENE, Shanghai, China) was used to introduce the mimic or the NC specifically into the lungs of each mouse before irradiation. The mouse model of RIPF was established using a protocol from our previous study [17]. Briefly, the lungs of mice were locally irradiated by 25 Gy of ^60^ Co γ-ray at a 200 cGy/min dosage rate, and the other parts of mice were shielded with 10 cm thick lead bricks. Five mice lung tissues were taken from each group at 1, 2, 3, and 4 months after irradiation. The lung tissues were removed for RNA and protein extraction, H&E staining, Masson staining, or immunohistochemical (IHC) staining. The animal experiments were approved by the Animal Care and Use Committee at the Military Academy of Medical Sciences, proceeded following the Laboratory Animal Guideline of Welfare and Ethics of China.

### 4.5. Western Blot Analysis and Antibodies

The Western bolt analysis was performed as described previously [18]. The protein concentrations of cell or tissue lysates were measured using the Bicinchoninic Acid Assay (BCA) (TIANGEN, Beijing, China, PA115). Afterward, 60 µg protein was separated using 10% sodium dodecyl sulfate polyacrylamide gel electrophoresis and electroblotted onto a nitrocellulose membrane. The membranes were blocked with 5% non-fat milk for two hours at room temperature, immunoblotted with specific primary antibodies. The antibodies used in this experiment were as follows: anti-E-cadherin (CST, Boston, MA, USA, 3195S; 1:1, 000); anti-N-cadherin (CST, Boston, MA, USA, 13116S, 1:1, 000); anti-Vimentin (Abcam, Cambridge, UK, ab8978, 1:1, 000); anti-Slug (Abcam, Cambridge, UK, ab51772, 1:1, 000); anti-β-actin (ZSGB-BIO, Beijing, China; TA-09, 1:1, 000); anti-GAPDH (Santa Cruz, CA, USA, sc-25778, 1:1, 000); anti-MZF1 (Santa Cruz, CA, USA, 293218, 1:1, 000). All antibodies were used following the manufacturer’s instructions. Protein expression was detected using a chemiluminescence agent (Thermo, Waltham, MA, USA). ImageJ software (Bethesda, MD, USA) was employed to quantify the results.

### 4.6. Immunofluorescence Analysis

A total of 2.5 × 10^5^ A549 cells were inoculated in six-well plates and transfected with siRNA/inhibitor or NC. Further, they were treated with 6 Gy of IR. The cells were washed three times with ice-cold PBS after 48 h and then fixed in 4% paraformaldehyde at room temperature for 30 min. The cells were permeabilized by treating with 0.3% Triton X-100 and washed before blocking. Then, cells were blocked in 10% FBS in PBS for 35 min at room temperature and incubated with anti-E-cad (CST, Boston, MA, USA, 3195S; 1:500) and anti-N-cad (CST, Boston, MA, USA, 13116S; 1:500) antibodies overnight at 4 °C. Afterward, the cells were incubated with the corresponding fluorescence-labeled secondary antibodies (Invitrogen; A21202/A11037; Thermo Fisher Scientific, Waltham, MA, USA), followed by blocking using a blocker containing DAPI (ZSGB-BIO, ZLI-9557, Beijing, China). The results were observed using X-LIGHT V3 (CRESTOPTICS, Rome, Italy) and NIKON TI2-E (Tokyo, Japan) capture system and quantified using the ImageJ software (Bethesda, MD, USA).

### 4.7. Dual-Luciferase Reporter Gene Assay

The JASPAR (http://jaspar.genereg.net/accessed on 18 October 2021) database was used for predicting the transcription factors regulating miR-541-5p. TargetScan (http://www.targetscan.org/vert_71/ accessed on 18 October 2021) database was used for predicting the target genes for miR-541-5p. The WT/MUT plasmids of the miR-541-5p promoter (cloned into PGL3-Basic) and the WT/MUT plasmids of Slug (cloned into pmirGLO) were transfected into HEK-293T cells together with MZF1 or miR-541-5p mimic. After 48 h, the dual-luciferase reporter gene assay was performed using the Dual-Luciferase Reporter Kit (Promega, San Luis Obispo, WI, USA). The fluorescence was measured using SpectraMax i3X (Molecular Devices, San Jose, CA, USA).

### 4.8. Hematoxylin and Eosin (H&E) and Masson’s Triple Stain

Lung tissues were fixed in 4% paraformaldehyde, embedded in paraffin, and cut into pathological sections. Lung fibrosis severity was detected by hematoxylin and eosin (H&E; ZSGB-BIO, Beijing, China; ZLI-9610) staining and quantitated by a semi-quantitative scoring system in Szapiel. Masson’s triple stain was performed by Masson’s Trichrome Stain Kit (Solarbio Life Science, G1340, Beijing, China). The images were acquired using Nikon’s Eclipse E600 research microscope (Nikon, Tokyo, Japan) and quantified using ImagePro Plus software (Bethesda, MD, USA).

### 4.9. Immunohistochemistry (IHC) Assay

For immunohistochemistry, sections were deparaffinized with xylene and rehydrated. Antigen retrieval was performed in 0.01 M citrate buffer (pH 6.0) using a pressure cooker for 2 min, followed by 3% hydrogen peroxide treatment for 5 min and washed with PBS. Specimens were incubated with primary antibody (Slug, Abcam, Cambridge, UK, ab51772, 1:200), overnight at 4 °C. The next day, tissue was washed with PBS and then treated with the corresponding secondary antibody for 1hour at room temperature. Subsequently, the tissue was dyed using 3,3N-Diaminobenzidine (DAB; ZSGB-BIO, Beijing, China; ZLI-9019) and hematoxylin solutions. The slides were sealed and examined under a microscope (Olympus, Tokyo, Japan) at 200× magnification. Each antigen was assigned an H-score. Briefly, H = ∑(pi*i), where “pi” denotes the percentage of positive cells and “i” denotes the intensity (weak intensity × 1, moderate intensity × 2, strong intensity × 3).

### 4.10. Statistical Analysis

All data were expressed as the mean ±SD. The differences were considered significant at *p* < 0.05. Unpaired numerical data were compared using the unpaired *t*-test (for comparison of two groups) or ANOVA (for comparison of over two groups). The data were analyzed using the SPSS software (IBM, Chicago, IL, USA).

## 5. Conclusions

The in vitro and in vivo experiments demonstrated the importance of miR-541-5p in the development of RIPF and that the response of miR-541-5p to irradiation is based on the activation of the transcriptional repressor MZF1. It was revealed that activated MZF1 induces the onset of EMT, which is an important link in the RIPF process, via the miR-541-5p/Slug axis, consequently accelerating the development of RIPF. In addition, the present study is the first to report changes in MZF1 upon irradiation. The present work provides a solid theoretical basis for the role of miR-541-5p as an important suppressor of RIPF.

## Figures and Tables

**Figure 1 ijms-22-11309-f001:**
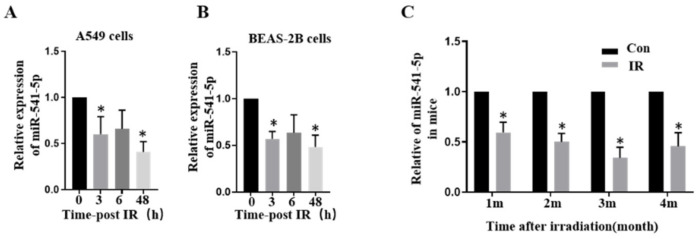
Ionizing radiation can reduce the expression of miR-541-5p. We used RT-qPCR at 0 h, 3 h, 6 h, and 48 h after 6Gy irradiation to detect miR-541-5p in A549 cells (**A**) and Beas 2B cells (**B**). (**C**) After 25Gy chest irradiation, lung tissues of mice were collected at the first, second, third, and fourth month, and the expression of miR-541-5p was detected by RT-qPCR. * *p* < 0.05 versus the control.

**Figure 2 ijms-22-11309-f002:**
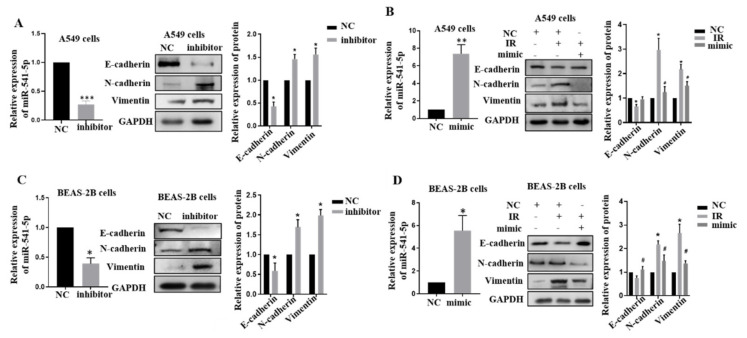
IR downregulated miR–541–5p to promote EMT. (**A**) Real-time PCR analysis verified the transfection efficiency of the miR–541–5p inhibitor (50 nm) in A549 cells. Detection of EMT-related protein changes after 48 h transfection by WB. The bar graph on the right shows the quantitative analysis of the protein using ImageJ. (**B**) Real-time PCR verified the transfection efficiency of the miR–541–5p mimic (50 nm) in A549 cells. Right: Re–overexpression of miR–541–5p after irradiation to observe the changes in the EMT-related proteins and the quantitative analysis of these proteins. Real-time PCR verified the transfection efficiency of the miR–541–5p inhibitor (**C**)/mimic (**D**) in Beas 2B cells. Detection of EMT–related protein changes after 48 h transfection by WB. The bar graph on the right shows the quantitative analysis of the protein using ImageJ. The bar graphs show the gray value analysis. * *p* < 0.05 versus the control; # *p* < 0.05 versus IR. ** *p* < 0.01, *** *p* < 0.001

**Figure 3 ijms-22-11309-f003:**
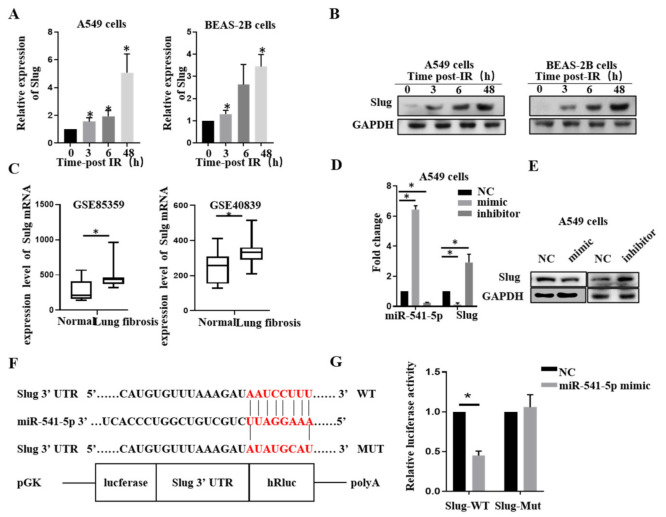
*Slug* was revealed as a direct target of miR-541-5p. A549 cells (left) and Beas 2B cells (right) were irradiated with 6 Gy of radiation. Detection of (**A**) *Slug* mRNA and (**B**) *Slug* protein using RT-qPCR or Western blotting at 0, 3, 6, and 48 h. (**C**) Comparison of *Slug* mRNA in lung tissues between a normal population and lung fibrosis patients (GSE40839) or mice (GSE85359) from NCBI (https://www.ncbi.nlm.nih.gov/ accessed on 18 October 2021) (**D**) Transfection of the miR-541-5p mimic and inhibitor in A549 cells and determination of the expression level of *Slug* mRNA using RT-qPCR. (**E**) Detection of the protein expression of *Slug* using Western blot analysis. (**F**) Information regarding the 3′-UTR binding site for the binding between miR-541-5p and Slug. (**G**) *Slug* 3′-UTR WT (wild-type)/MUT (mutant) and the miR-541-5p mimic cotransfected in HEK-293T cells, followed by the detection of luminescence based on a dual-luciferase reporter system. * *p* < 0.05 versus the control.

**Figure 4 ijms-22-11309-f004:**
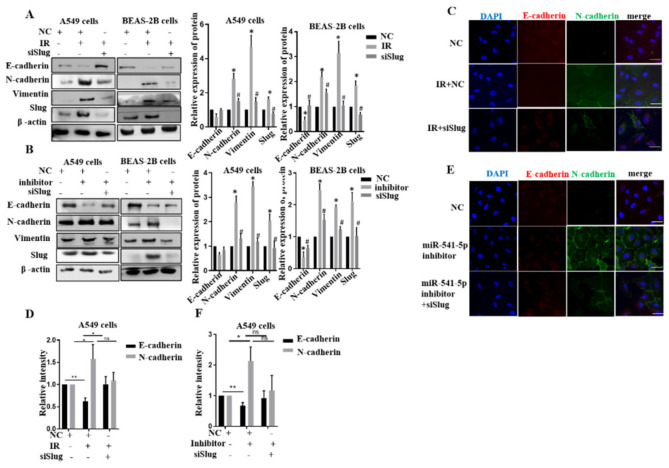
The siSlug inhibited the EMT induced by IR combined with the knockdown of miR–541–5p. (**A**) Transfection of siSlug (100 nM) after irradiation of A549 and BEAS-2B cells using 6 Gy and detection of EMT-related protein expression after 48 h. The histogram shows the gray value analysis. (**B**) A549 and Beas 2B cells cotransfected with miR–541–5p inhibitor and siSlug. Left: the expression of the EMT–related proteins was examined 48 h after switching to the normal medium; Right: the quantitative analysis of the protein changes using ImageJ. (**C**) Immunofluorescence analysis examined the expression of the EMT–related proteins E-cadherin and N–cadherin through the transfection of siSlug after 6 Gy irradiation. Scale bar, 20 µm (**D**) The bar graph was generated by quantitatively analyzing the expression area and the relative fluorescence using ImageJ. * *p* < 0.05, ** *p* < 0.01 (**E**) Co–transfection of miR–541–5p and siSlug in A549 cells and immunofluorescence detection to determine the expressions of E–cadherin and N–cadherin in the cells. Scale bar, 20µm (**F**) The bar graph presents the quantified immunofluorescence results obtained using ImageJ. Data represents the mean ± SEM (*n* = 3), * *p* < 0.05 versus control of the same group; # *p* < 0.05 versus IR or inhibitor of the same group. ns: no significant.

**Figure 5 ijms-22-11309-f005:**
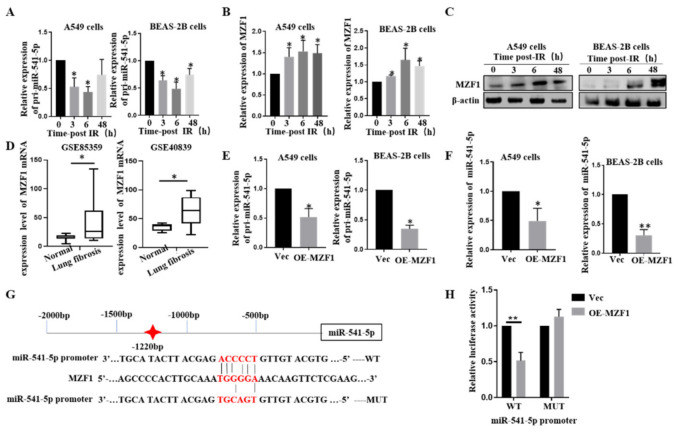
The transcription factor *MZF1* regulated the expression of miR–541–5p upon IR. (**A**) The expression of pri–miR–541–5p was detected using RT–qPCR at 0, 3, 6, and 48 h after irradiation. Changes in the expression of (**B**) *MZF1* mRNA and (**C**) *MZF1* protein in A549 and BEAS–2B cells at 0, 3, 6, and 48 h after 6 Gy irradiation, detected using RT–qPCR and Western blot analysis. (**D**) Comparison of *MZF1* mRNA in lung tissues between a normal population and lung fibrosis patients (GSE40839) or mice (GSE85359) from NCBI (https://www.ncbi.nlm.nih.gov/ accessed on 18 October 2021). Overexpression of *MZF1* in A549 cells and BEAS–2B cells followed by detection of (**E**) pri–miR–541–5p and (**F**) miR–541–5p using RT–qPCR. (**G**) Predicting the binding site for the binding between *MZF1* and the miR–541–5p promoter region using the JASPAR database. (**H**) Co–transfection of the miR–541–5p promoter WT (wild type)/MUT (mutant) and the overexpression of *MZF1* plasmid in HEK-293T cells, followed by the detection of luminescence based on a dual-luciferase reporter system. * *p* < 0.05, ** *p* < 0.01.

**Figure 6 ijms-22-11309-f006:**
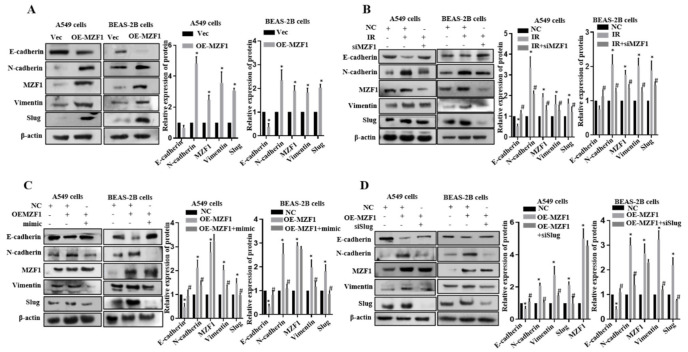
*MZF1* promoted EMT by repressing the miR–541–5p expression upon IR. (**A**) Overexpression of *MZF1* in A549 cells and BEAS–2B cells to observe changes in EMT–related protein levels. The bar graph presents the results of the gray value analysis. (**B**) Western blot experiments were performed to verify whether the IR-induced changes in the EMT–related proteins were inhibited upon the knockdown of *MZF1* in A549 cells and Beas 2B cells. The bar graph presents the results of the gray value analysis. * *p* < 0.05 versus the control; # *p* < 0.05 versus IR. (**C**) Co–transfection of *MZF1* with the miR–541–5p mimic in cells to observe the changes of the EMT–related proteins and target genes. (**D**) Co–transfection of *MZF1* with siSlug in cells to determine the EMT–related protein. Data represents the mean ± SEM (*n* = 3), * *p* < 0.05 versus control of the same group; # *p* < 0.05 versus OE-MZF1 of the same group.

**Figure 7 ijms-22-11309-f007:**
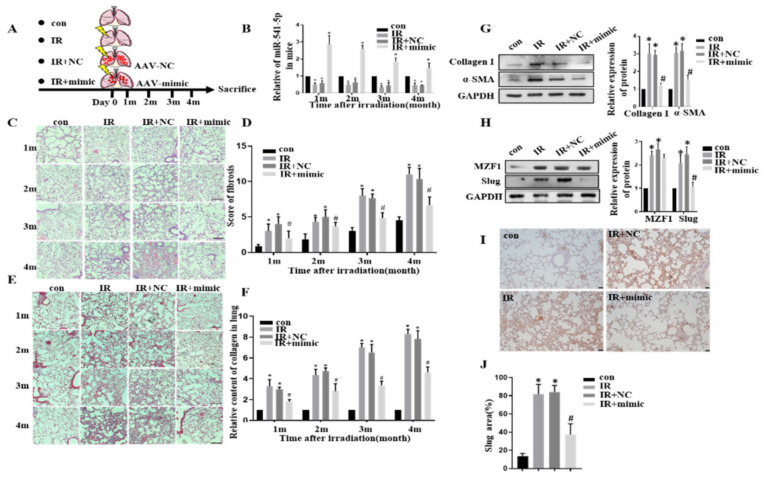
MZF1 mediated the radiation-induced pulmonary fibrosis through the miR-541-5p/Slug axis. (**A**) Diagram depicting the mice grouping. (**B**) Detection of miR–541–5p in mouse lung tissue using RT–qPCR. (**C**,**D**) H&E staining and scoring of the mouse lung tissue sections (using the semi-quantitative method of lung pathology). The scale bar represents 100 µm. (**E**,**F**) Masson’s staining and quantitative analysis of the mouse lung tissue sections (ImagePro Plus). (**G**) Detection of Collagen and α–SMA expression in mouse lung tissue using the Western blot assay. (**H**) MZF1 and Slug expressions in mouse lung tissue determined using the Western blot assay. (I) Slug expression in mouse lung tissue by immunohistochemical experiment. The scale bar represents 10 µm. (**J**) Quantitative analysis of IHC staining. Data represents the mean ± SEM (*n* = 5), * *p* < 0.05 versus NC of the same group; # *p* < 0.05 versus IR + NC.

**Figure 8 ijms-22-11309-f008:**
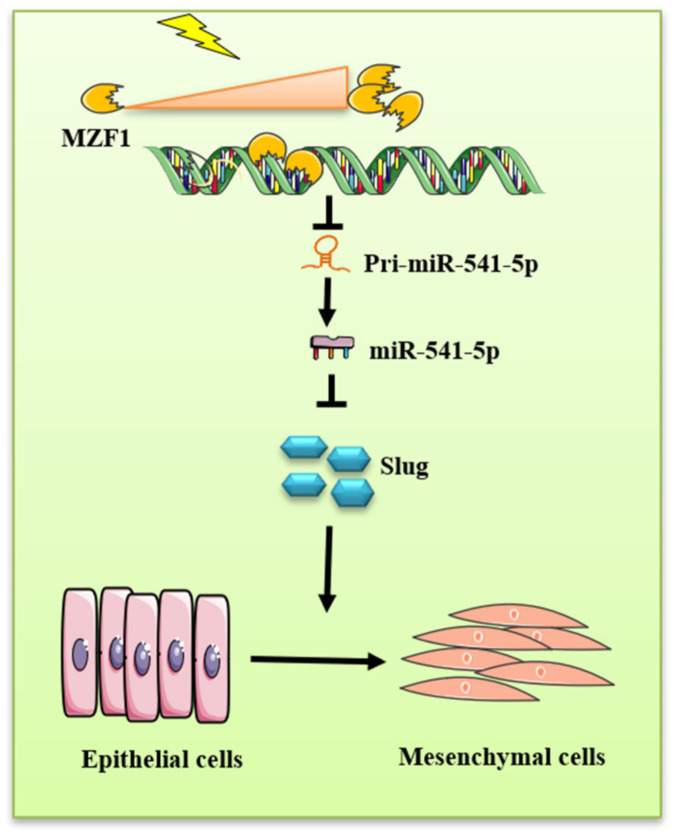
Mechanistic diagram illustrating the functioning of miR-541-5p as a promoter of EMT under IR conditions. IR activated the transcription factor *MZF1*, thereby increasing its levels. The increased *MZF1* could bind to the promoter region of miR-541-5p, thereby inhibiting the production of pri-miR-541-5p, which consequently reduced the levels of miR-541-5p. With the reduction in the miR-541-5p levels, the silencing effect on the downstream gene *Slug* was diminished, which consequently induced the EMT process.

**Table 1 ijms-22-11309-t001:** RT-qPCR primers.

Primer ID	Sequence (5′-3′)
Has-miR-541-5p-F	AGGATTCTGCTGTCGGT
Has-miR-541-5p-R	GGTCCAGTTTTTTTTTTTTTTTAGTG
Slug-F	GACTGACCCGTCGTGACG
Slug-R	GCAGACGACGGGTCAGAT
Pri-miR-541-5p-F	ACGGTGCATGTCATCTGTTC
Pri-miR-541-5p-R	AAGATGTCACAGACGACTTC
MZF1-F	GGGCCTGCAGGTGAAAGAG
MZF1-R	GGCAGCTAGAGGCCCAGACT
Has-U6-F	ATTGGAACGATACAGAGAAGAAT
Has-U6-R	GGAACGCTTCACGAATTTG
β-actin-F	GAATCAATGCAAGTTCGGTTCC
β-actin-R	TCATCTCCGCTATTAGCTCCG

## Data Availability

Not applicable.

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
