# Peer review of "Irradiation Activates MZF1 to Inhibit miR-541-5p Expression and Promote Epithelial-Mesenchymal Transition (EMT) in Radiation-Induced Pulmonary Fibrosis (RIPF) by Upregulating Slug"

_ijms, 2021, doi:10.3390/ijms222111309_

Round 1

Reviewer 1 Report

This article describes a mechanistic relationship between MZF1 transcription factor high expression induced by irradiation and Slug upregulation, which is associated to radiation induced pulmonary fibrosis (RIPF) through promoting EMT in alveolar epithelium. This relationship would be mediated by miRNA miR-541-5p, which is a negative regulator of Slug, and whose expression would be downreglulated by MZF1. The authors show an array of experiments which show convincingly this mechanism, and, additionally, point towards miR-541-5p as a lever to modulate the RIPF

These results could be of great value for patients under chest irradiation oncologic treatment as far as they suggest a potential way to mitigate secondary effects of these therapies.

I consider this work deserves publication, but I would like to do some remarks: 

  • When authors describe and discuss the results of the experiments in sections 2.2, 2.4, 2.6, they conclude that EMT process is reversed by their experimental manipulations. However, I think this claim is not completelly justified as far as the set up of the experiments only show that the EMT process do not reach a full development compared to the control groups. I mean that the EMT process would be reversed if it would have been allowed to fully develop and then a treatment would have led things to an initial stage. In fact, their experiments show a partial negative modulation of the EMT process  and a retardation in the RIPF onset and development.
  • The dual-luciferase reporter gene assay shows that MZF1 regulates negativelly miR-541-5p expression. However, the interaction between the MZF1 transcription factor and the DNA secuence could be indirect. A direct interaction could be assessed by performing an EMSA assay. Have the authors thougt of this kind of assay?

Other minor points that should be revised in the final version of the paper are:

  • Legends for Figures 2, 4, and 6 refer to bar graphs as histograms. This should be corrected.
  • References format do not adapt to journal instructions (Ref. nr. Author 1, A.B.; Author 2, C.D. Title of the article. Abbreviated Journal Name Year, Volume, page range).

Author Response

Dear Reviewer:

Thank you for your time and nice comments on our article. Your comments are all of great impirtance to our article. Based on the comments, we attached a point-by-point letter.Please see the attachment.

Sincerely,

Yongqing Gu,

Reviewer 2 Report

The work of Xinxin Liang and colleagues is aimed at establishing the regulation axis in development of radiation-induced pulmonary fibrosis (RIPF) implemented through miR-541-5p and Slug transcription factor. The work is well designed and performed at high methodological standard. The authors presented interesting results supporting miR-541-5p role in IR-induced EMT and subsequent fibrosis and demonstrated possibility to prevent fibrosis development by AAV-based vaccine. The latter finding is of particular practical importance. To my opinion, the work can be published after minor revision.

Minors

  • English must be revised. There are many errors and typos in the manuscript. The examples are: extra words in Line 33; inconsistent sentencein Lines 41-42; using ‘by’ instead of ‘of’ in Line 54 and so on. Please, thoroughly revise the text.
  • Please, decipher all the acronyms at first mentioning, including those appearing in Discussion and Methods sections.
  • Line 145. Is it knockdown or knockout?
  • 4 D,F. I recommend using the term ‘relative fluorescence’, not just intensity. Line 162. Is it miR-541-5p or miR-541-5p inhibitor?
  • Line 177. It is not ‘the promoter site of miR-541-5p’ itself.
  • 6 capture. Now, the meaning of symbols used to show statistical significance is explained twice. Is it necessary?
  • Line 264. What do the authors mean under ‘prolonged irradiation time’?
  • Lines 323-326. This is the information about AECII cells response to IR; but what is the source for the data?
  • Lines 363-364. The reference must be added.
  • The Materials and Methods section is the less accurate section in the manuscript in its present form. Please, provide all the necessary details to make the study reproducible. Please, provide the cat numbers for all the purchased RNAs or the link to the source with their sequences. Please, correct the ’50 nm’ in Line 416. All the reagents must be described with full names and manufacturers (lip2000, for example). In WB description, no information is provided about beta-actin Abs. Line 445, the phrases like ‘a certain number of cells’ should be avoided. Line 446, it it incorrect description of the experimental procedure. The manufacturers must be provided for all the equipment and software. There is no information about Abs in IHC section. This list of inaccuracies in the section is incomplete, so, I ask the authors to thoroughly revise it.
  • There are no names of journals in the list of references.

Author Response

Dear Reviewer:

Thank you for your time and nice comments on our article. Your comments are all of great impirtance to our article. Based on the comments, we attached a point-by-point letter. Please see the attachment.

Sincerely,

Yongqing Gu
